# Immunotherapy for Esophageal Cancers: What Is Practice Changing in 2021?

**DOI:** 10.3390/cancers13184632

**Published:** 2021-09-15

**Authors:** Hannah Christina Puhr, Matthias Preusser, Aysegül Ilhan-Mutlu

**Affiliations:** 1Division of Oncology, Department of Medicine I, Medical University of Vienna, 1090 Vienna, Austria; hannah.puhr@meduniwien.ac.at (H.C.P.); Matthias.Preusser@meduniwien.ac.at (M.P.); 2Comprehensive Cancer Center Vienna, 1090 Vienna, Austria

**Keywords:** esophageal cancer, esophageal adenocarcinoma, esophageal squamous cell carcinoma, immunotherapy

## Abstract

**Simple Summary:**

Recent studies concerning immunotherapeutic agents are bound to revolutionize therapeutic strategies for esophageal cancer patients. This review aims to summarize novel clinical data and provide a critical view of potential restrictions for the implementation of immunotherapy for unselected patient populations.

**Abstract:**

The prognosis of advanced esophageal cancer is dismal, and treatment options are limited. Since the first promising data on second-line treatment with checkpoint inhibitors in esophageal cancer patients were published, immunotherapy was surmised to change the face of modern cancer treatment. Recently, several studies have found this to be true, as the checkpoint inhibitors nivolumab and pembrolizumab have achieved revolutionary response rates in advanced as well as resectable settings in esophageal cancer patients. Although the current results of large clinical trials promise high efficacy with tolerable toxicity, desirable survival rates, and sustained quality of life, some concerns remain. This review aims to summarize the novel clinical data on immunotherapeutic agents for esophageal cancer and provide a critical view of potential restrictions for the implementation of these therapies for unselected patient populations.

## 1. Introduction

Cancer of the esophagus and the gastric esophageal junction (GEJ) is a devastating disease and a major contributor to the global disease burden [1]. In 2020, esophageal cancer was declared to be the seventh-most common oncological disease worldwide by the GLOBOCAN 2020 estimates, with 604,100 (3.1% of all sites) newly diagnosed cases, and the sixth-most common cancer causing mortality, with 544,076 new deaths (5.5% of all sites) [2]. Although the incidence in Western regions is quite low [1], this cancer poses a major global health issue, with epidemiological hot spots in Asia and Africa [3,4]. In addition to this unique epidemiological feature of an uneven geographical distribution, there are also variations in ethnicity and gender, with higher risk for males as well as African and Asian ethnicities [1]. In recent years, it became evident that these differences might be based on diverse tumor biology and, thus, may influence treatment response and outcome.

Furthermore, although this malignant disease is often described as one entity based on its anatomical location, there are vast differences between the two predominant histological subtypes, esophageal squamous cell carcinoma (SCC), which accounts for 80% of esophageal cancer cases, and esophageal adenocarcinoma (AC) [5]. However, throughout the past decades, the incidence of SCC, which comprises over 90% of all esophageal cancer cases in lower-income countries, is decreasing. While this phenomenon is surmised to be preceded by economic gains and dietary improvements in Asian regions, the reductions are considered primarily due to a decline in cigarette smoking in Western countries [2]. Thus, heavy drinking and smoking are considered to be major risk factors for SCC in high-income countries, while dietary components such as nutritional deficiencies, consumption of pickled vegetables, and hot food, as well as betel quid chewing, are surmised to increase the risk of cancer development in lower-income countries [2].

In addition, almost two-thirds of esophageal cancer cases in Western regions are ACs, and the incidence is rapidly increasing due to excess body weight, gastroesophageal reflux disease, and Barrett’s esophagus as the key risk factors [2]. As these trends are predicted to continue in the near future, it is surmised that AC might surpass SCC in many high-income countries within the next decades. Hence, the incidence as well as risk factors differ vastly in SCC and AC esophageal cancer patients. Recent analyses of the Cancer Genome Atlas Research Network even state that esophageal SCC resembles SCC of other organs more than it does esophageal AC [6]. Hence, large studies also focus on this issue in patient selection and frequently define histology or geographic region as the stratification parameter. Furthermore, the location of the tumor is also a viable factor, as most ACs occur in the lower thoracic esophagus and the GEJ [7], while SCCs often affect the middle and upper thoracic esophagus and might sometimes occur simultaneously with head and neck cancer [8]. This leads to a substantial variation in treatment options, depending on the histological subtype as well as tumor location.

However, therapeutic options are limited, independent of epidemiology, localization, or histological subtypes, and prognosis remains poor with 5-year survival rates around 5% in advanced stages [1]. New treatment options are therefore desperately needed to improve patient management. Since immunotherapy has proved to be effective in the treatment of various cancer entities, including head and neck and lung cancers, it might pose a viable treatment option for esophageal cancer as well [9]. Particularly, the success of immunotherapy in head and neck cancer leads to higher expectations for esophageal cancer due to similarity in the biological behavior of these two entities. This short review aims to summarize and critically review recent (potentially) practice-changing studies concerning immunotherapeutic agents in various subgroups of esophageal cancer patients. Table 1 and Table 2 give an overview of recent clinical trials discussed in this review, while Table 3 focuses on the approval status of immunotherapeutic agents by the US Food and Drug Administration (FDA) and the European Medicines Agency (EMA).

## 2. Esophageal Cancer in Localized Setting

### 2.1. Esophageal Adenocarcinoma

To discuss the current treatment highlights of localized esophageal cancer, first we have to shed some light on the current controversies of the therapeutic strategies of localized esophageal AC.

In 2012, the results of the CROSS trial presented a major breakthrough due to improved survival with preoperative chemoradiotherapy among patients with potentially curable esophageal or gastroesophageal junction cancer (included patients: 275 (75%) patients with AC, 84 (23%) patients with SCC, and 7 (2%) patients with large-cell undifferentiated carcinoma) [10]. Recently published data on 10-year-survival outcomes (median follow-up 147 months) affirm the overall survival benefit for patients receiving neoadjuvant chemoradiotherapy compared to surgery alone (HR 0.70; 95% CI 0.55–0.89; overall OS benefit 13% (38% versus 25%, respectively)) [11].

Another approach to improve the outcome in patients with localized gastroesophageal cancer is the docetaxel-based triplet FLOT regimen (fluorouracil plus leucovorin, oxaliplatin, and docetaxel), which shows improved OS compared to the former standard-of-care perioperative anthracycline-containing chemotherapy regimen recommended by the MAGIC trial (FLOT arm: 356 patients with AC, median OS 50 months (38.33–not reached) versus ECF/ECX arm: 360 patients with AC, median OS 35 months (27.35–46.26); HR 0.77; 95% CI 0.63–0.94) [12].

It is evident that neoadjuvant management (both chemotherapy and chemoradiation) in localized esophageal cancer patients is necessary and extends survival. However, as the results of both treatment strategies, CROSS and FLOT, yielded similar survival outcomes for patients with resectable esophageal cancer, there is currently no official tendency toward one or the other. However, several studies aim to identify the most promising path for patient management.

Recently presented first results of the phase III Neo-AEGIS trial, which compared the CROSS regimen with perioperative chemotherapy (both anthracycline- and latterly taxane-based regimen) in European patients with esophageal AC, revealed no evidence that perioperative chemotherapy is unacceptably inferior to multimodal therapy (3-year estimated survival probability of 56% (95% CI 47–64) and 57% (95% CI 48–65); HR 1.02; 95% CI 0.74–1.42) [13]. However, a major limitation of this analysis was the chosen chemotherapy regimen. Of the 362 randomly assigned evaluable patients, 184 received perioperative chemotherapy, yet only 27 (15% of the chemotherapy arm) received the currently recommended FLOT regimen. The reason for this inconsistency was the extended study period and inclusion of patients before the recommendation to administer a taxane-based rather than an anthracycline-based regimen. Although patients were treated according to the standard of care at the time of inclusion, the findings of this analysis, thus, are limited in the scope of current treatment options. Furthermore, the trial showed that the pathologic complete response is superior with the CROSS regimen than with perioperative chemotherapy (16% versus 5%, respectively) [13]. Nevertheless, this accomplishment could not be translated in survival benefit. Although no data on the metastatic pattern were available, it is surmised that this effect might be due to ineffective prevention of distant metastases by local multimodal therapy.

In addition, these data might once again be biased by the choice of chemotherapeutic regimen, as FLOT is known to achieve more extended complete response rates than an anthracycline-based regimen (16% vs. 6%, respectively) [14,15]. Although a cross-trial comparison has several limitations, previous FLOT trial data and Neo-AEGIS data demonstrate similar pathological complete response rates of CROSS and FLOT regimens.

Furthermore, a recently published propensity score-matched analysis comparing the postsurgical survival (30-day/90-day/1-year mortality), treatment response, and surgical complications of patients with AC who received either the CROSS (339 patients) or the FLOT protocol (97 patients) showed no significant differences between both groups [16]. However, so far there are no results of large randomized trials to compare neoadjuvant chemoradiotherapy as used in the CROSS regimen and taxane-based perioperative chemotherapy as used in the FLOT trial in terms of overall survival. Thus, the findings of the ongoing phase III ESOPEC trial are highly awaited to shed some light on this matter [17]. The results of this analysis will also provide further information about the toxicity of both regimens, which might be included in future treatment recommendations. Until then, both perioperative chemotherapy and neoadjuvant chemoradiotherapy are valid options according to international guidelines and should carefully be offered to patients after critical discussion by multidisciplinary tumor boards.

To further improve the response and outcome in patients with localized esophageal AC, the enhancement of these therapeutic strategies is of great scientific and clinical interest. Thus, the combination of currently available treatment strategies with new therapeutic drugs, such as immunotherapeutic agents, is considered highly promising.

The recently published global, randomized, placebo-controlled phase III CheckMate 577 trial investigated the effects of the checkpoint inhibitor nivolumab, which causes immune checkpoint blockade by diminishing inhibitory signaling through the programmed death receptor-1 (PD-1) pathway, as an adjuvant treatment option in patients with residual pathological disease (≥ypT1 or ≥ypN1) after neoadjuvant chemoradiation therapy. The adjuvant treatment with nivolumab improved the median disease-free survival (22.4 months; 95% CI 16.6–34.0) compared to the placebo (11.0 months; 95% CI 8.3–14.3; HR 0.69; 96.4% CI 0.56–0.86; *p* < 0.001) [18]. This benefit seemed to be independent of the histological subtype (AC: HR 0.75, 95% CI 0.59–0.96; SCC: HR 0.61, 95% CI 0.42–0.88) and PD-L1 expression (tumor-cell PD-L1 expression ≥1%: HR 0.75, 95% CI 0.45–1.24; <1%: HR 0.73, 95% CI 0.57–0.92; indeterminate or not evaluated: HR 0.54, 95% CI 0.27–1.05). At the American Society for Clinical Oncology (ASCO) 2021 congress, further data on metastasis-free survival (28.3 versus 17.6 months; HR 0.74; 95% CI 0.60–0.92) were presented, which clearly favored nivolumab over the placebo. Quality-of-life assessments demonstrated similar trends in improvement from baseline during treatment and maintained benefits post-treatment between nivolumab and the placebo, advocating nivolumab as a reliable treatment option without compromising the quality of life [19]. Thus, in May 2021, adjuvant nivolumab was approved by the Food and Drug Administration (FDA) for patients with completely resected esophageal or gastroesophageal junction cancer with residual pathologic disease who have received neoadjuvant chemoradiotherapy [20]. Approval by the European authorities was given in August 2021 for the same indication [21].

As a future concept for the improvement of perioperative chemotherapeutic strategies, the KEYNOTE-585 study was designed to evaluate the efficacy and safety of pembrolizumab plus chemotherapy compared with the placebo plus chemotherapy as neoadjuvant/adjuvant treatment for localized gastroesophageal junction AC, and the first results are awaited to be published in the near future [22].

### 2.2. Esophageal Squamous Cell Carcinoma

Current treatment options for localized esophageal SCC partially overlap with recommendations for AC. However, as discussed at the beginning of this review, there are vast differences in tumor development and therefore treatment response. Although only one quarter of the treatment population’s histological subtype of the CROSS study was represented by SCC, multimodal therapy seemed to have the largest impact on this subgroup. With a highly significant pathological complete response of 49% (versus 23% in AC patients; *p* = 0.008) and estimated 5-year survival benefit from combination chemoradiotherapy compared to surgery alone (AC: *p* = 0.049; SCC: *p* = 0.011), it is evident that this patient subgroup had a major benefit from this therapeutic intervention [10]. However, the histologic tumor type was not a prognostic factor for survival, which indicates that both tumor subtypes benefited from neoadjuvant chemoradiotherapy, and thus, the CROSS regimen is currently recommended for both patient populations.

In contrast, the combination of the CROSS regimen with adjuvant nivolumab in the aforementioned CheckMate 577 trial showed a greater benefit in disease-free survival in the SCC subgroup as well (AC: placebo 11.1 months (95% CI 8.3–16.8) versus nivolumab 19.4 months (95% CI 15.9–29.4); SCC: placebo 11.0 months (95% CI 7.6–17.8) versus nivolumab 29.7 months (95%CI 14.4–not estimated)). These findings indicate that the combination of systemic and radiotherapy acts by sensitizing SCC cells and thereby leads to a major survival benefit compared to other strategies.

Due to the high complete pathological response rates of almost 50% and high efficacy of the combined chemoradiation therapy regimen, this treatment approach is even used without surgical resection as a curative attempt in particularly irresectable localized tumors, in tumors with a cervical location, and for inoperable patients [23]. In this scenario of the so-called definitive radiochemotherapy, the doses of radiation therapy are usually extended to a higher volume in order to achieve better local control [24]. Whether this patient cohort also benefits from the addition of an immunotherapy agent is not clear due to lacking data. Thus, to further improve the survival of these patients, several trials addressing this issue are currently ongoing. The KEYNOTE-975 trial aims to investigate the benefit of the combination of definitive chemoradiotherapy plus pembrolizumab in patients with esophageal carcinoma [25]. Furthermore, the SKYSCRAPER-07 trial evaluates the efficacy and safety of tiragolumab, a monoclonal antibody designed to bind with TIGIT (a protein receptor on immune cells), plus atezolizumab, a humanized monoclonal antibody immune checkpoint inhibitor that selectively binds to PD-L1, compared with the placebo in participants with localized but unresectable esophageal SCC following definitive concurrent chemoradiotherapy. The results of these trials are awaited in order to change the therapeutic approaches in this patient population with definitive chemoradiotherapy.

In addition, there are also novel immunotherapeutic strategies to improve neoadjuvant treatment strategies. The Henan Cancer Hospital Thoracic Oncology Group 1909 (HCHTOG1909), for example, initiated a phase III study on neoadjuvant chemotherapy versus neoadjuvant toripalimab, a humanized PD-1 monoclonal antibody, plus chemotherapy for locally advanced esophageal SCC [26].

## 3. Esophageal Cancer in Advanced and Metastatic Settings

### 3.1. Esophageal Adenocarcinoma

With regard to esophageal AC in advanced settings, several practice-changing trials were recently published. The phase III CheckMate 649 study revolutionized the first-line treatment of esophageal AC by showing that the addition of nivolumab to standard chemotherapy leads to an improvement of the OS (HR 0.71; 98.4% CI 0.59–0.86; *p* < 0.0001) as well as progression-free survival (PFS) (HR 0.68; 98% CI 0.56–0.81; *p* < 0.0001) versus chemotherapy alone in patients with a PD-L1 combined positive score (CPS) of 5 or more (nivolumab + chemotherapy arm: primary tumor location, 333 (70%) gastric, 84 (18%) GEJ, 56 (12%) esophageal cancer patients; race: 119 (25%) Asian, 328 (69%) white, American Indian, or Alaska native 10 (2%), black or African American 2 (<1%), and others 14 (3%); chemotherapy arm: primary tumor location: 334 (69%) gastric, 86 (18%) GEJ, 62 (13%) esophageal cancer patients; race: 117 (24%) Asian, 327 (68%) white, 10 (2%) American Indian or Alaska native, 7 (1%) black or African American, 21 (4%), and others) [27]. Furthermore, recent data showed maintained tolerability as well as quality of life, providing further support for this combination to be the first-line treatment of choice in this patient cohort [28]. Thus, this new therapeutic strategy was approved by the FDA, independent of PD-L1 expression (although the greatest benefit was shown for PD-L1-positive tumors) [29].

Furthermore, pembrolizumab in combination with chemotherapy was recently approved by the FDA as a first-line option for patients with advanced and metastatic esophageal cancer independent of histological subtype and PD-L1 status based on the results of the KEYNOTE-590 trial (all patients: pembrolizumab + chemotherapy median OS 12.4 (*n* = 373) versus placebo + chemotherapy 9.8 (*n* = 376) months; HR 0.73; 95% CI 0.62–0.86; *p* < 0.0001) [30,31]. The EMA, however, placed a restriction on the decision, as treatment with pembrolizumab was approved only for patients with a PD-L1 CPS of ≥10 (esophageal squamous cell carcinoma and PD-L1 CPS of ≥10: median OS 13.9 (*n* = 143) versus 8.8 (*n* = 143) months; HR 0.57; 95% CI 0.43–0.75; *p* < 0.0001; esophageal squamous cell carcinoma: 12.6 (*n* = 274) versus 9.8 (*n* = 274) months; HR 0.72; 95% CI 0.60–0.88; *p* = 0.0006; PD-L1 CPS ≥10: 13.5 (*n* = 186) versus 9.4 (*n* = 197) months; HR 0.62; 95% CI 0.49–0.78; *p* < 0.0001) [32,33]. Both authorities defined the chemotherapy backbone as platine and fluoropyrimidine based and did not make any clear statement on the specific choice of backbone treatment combination for the KEYNOTE-590 regimen.

However, when discussing advanced AC of the esophagus, it is important to not only address the biomarkers for immunotherapeutic agents but also further investigate other biomarkers. Human epidermal growth factor receptor 2 (HER2) is a known marker for pathogenesis and poor outcomes in several tumor entities. However, the 2010 published ToGA trial led to a vast improvement in survival in this patient subgroup due to the combination of trastuzumab, a monoclonal antibody against HER2, with chemotherapy in a first-line setting (trastuzumab + chemotherapy (236 [80%] gastric, 58 [20%] GEJ cancer patients) median OS 13.8 months (95% CI 12–16) versus chemotherapy alone (242 [83%] gastric, 48 [17%] GEJ cancer patients) 11.1 months (10–13) in those assigned to chemotherapy alone; HR 0.74; 95% CI 0.60–0.91; *p* = 0.0046) [34]. Although this trial only included AC of the gastric or gastro-esophageal junction (20%) and no esophageal cancer, patients with HER2-positive AC of the esophagus are used to be treated based on the data of this trial in many oncological centers. Consistently, a recently published investigator-initiated phase II trial investigating the addition of pembrolizumab to the modified ToGA regimen (cisplatin/oxaliplatin plus capecitabine/5-fluorouracil plus trastuzumab) in HER2-positive gastroesophageal cancer patients included 15 patients (38% of the study population) with esophageal AC. The results of this trial were promising concerning safety and efficacy (26 (70%; 95% CI 54–83) of 37 patients were progression free at 6 months) [35]. Thus, the large phase III KEYNOTE-811 trial was initiated. The first results of this study were presented at the ASCO 2021 meeting and showed a significant benefit for the overall response rate (ORR) (pembrolizumab + ToGA: ORR 74.4%; 95% CI 66.2–81.6; placebo + ToGA: ORR 51.9%; 95% CI 43.0–60.7; *p* = 0.00006) with this new combination of immunotherapy with targeted therapy and chemotherapy [36]. Although, further results on the outcome are still missing and expected to be published in the near future, the FDA has already approved this treatment strategy [37]. It is, however, important to mention that despite a significant number of patients with esophageal location within the initial phase II trial, the phase III KEYNOTE-811 trial did not include esophageal AC patients and, thus, the approval of the FDA was restricted to gastric and GEJ locations. It is, however, assumable that this regimen will soon be implemented in clinical routine in esophageal AC patients as well, just as it was the case with the ToGA regimen.

### 3.2. Esophageal Squamous Cell Carcinoma

In advanced SCC, recent advances in immunotherapeutic strategies were achieved in first-line settings, as mentioned in the KEYNOTE-590 trial above, as well as in further-line settings. The ATTRACTION-3 trial showed a significantly improved OS in the nivolumab group compared with the chemotherapy group as second-line treatment in advanced or metastatic esophageal SCC (10.9 months, 95% CI 9.2–13.3 versus 8.4 months, 95% CI 7.2–9.9; HR 0.77, 95% CI 0.62–0.96; *p* = 0.019). Although 96% of the study population consisted of patients with Asian ethnicity, both the FDA and the EMA approved nivolumab as a second- and further-line therapy for patients with unresectable advanced, recurrent, or metastatic esophageal SCC [38,39,40]. Furthermore, the phase II KEYNOTE-180 study, which evaluated pembrolizumab in a third and further- setting in advanced and metastatic esophageal cancer independent of histological subtype, showed a promising ORR of 14.3% (95% CI 6.7–25.4) among patients with SCC and 5.2% (95% CI 1.1–14.4) among patients with AC [41]. Based on these positive results, the KEYNOTE-181 trial, investigating pembrolizumab versus chemotherapy in patients with advanced/metastatic SCC or AC of the esophagus, which progressed after one prior therapy session, was initiated. Although pembrolizumab showed promising results in the overall cohort, the most significant benefit was seen in SCC (8.2 versus 7.1 months; HR 0.78; 95% CI 0.63–0.96; *p* = 0.0095) and a PD-L1 CPS of ≥10 (9.3 versus 6.7 months; HR 0.69; 95% CI 0.52–0.93; *p* = 0.0074) [42]. Thus, pembrolizumab was only approved in this patient subcohort (SCC + PD-L1 CPS ≥ 10) as a second- and further-line therapy [43].

Based on the results of the ATTRACTION-3 trial, nivolumab was also explored in metastatic esophageal squamous cell carcinomas as a therapeutic option in a first-line setting. The CheckMate 648 trial randomized patients into one of these three arms: (i) the combination of nivolumab plus ipilimumab, an anti-CTLA-4 (cytotoxic T-lymphocyte-associated protein 4) antibody, (ii) nivolumab plus chemotherapy, or (iii) chemotherapy alone. Recently presented results show that both nivolumab plus ipilumumab (median OS 13.7 months; HR 0.64; 98.6% CI 0.46–0.90; *p* = 0.001) and nivolumab plus chemotherapy (median OS 15.4 months; HR 0.54; 99.5% CI 0.37–0.80; *p* < 0.0001) are superior to chemotherapy alone (median OS 9.1 months) in patients with tumor cell PD-L1 ≥ 1% [44]. There was some benefit for PD-L1-negative patients in both experimental groups compared to chemotherapy alone, and longer follow-up will help us decide the extent of this benefit. In the nivolumab-plus-ipilimumab arm, there was a cross-over of the survival curves, indicating rapidly progressing patients within the first months of treatment under this chemotherapy-free regimen. Further subgroup analyses might help us select patients who truly benefit and do not experience additional harm from this regimen. However, so far no further approval statement has been published by the FDA.

Other potential targets that were presented at the ASCO 2021 meeting as promising new therapeutic options are the humanized anti-PD-1 monoclonal antibodies camrelizumab and tislelizumab. Camrelizumab was investigated in the ESCORT-1st trial in combination with chemotherapy in a first-line setting in metastatic esophageal SCC in a Chinese cohort [45]. The combination with immunotherapy led to an improved OS compared to chemotherapy with the placebo (15.3 versus 12.0 months; HR 0.70; 95% CI 0.56–0.88; one-sided *p* = 0.0010) irrespective of PD-L1 expression. Based on this trial, the study group seeks approval from the China National Medical Products Administration for camrelizumab plus chemotherapy in the treatment of untreated advanced or metastatic esophageal SCC. Camrelizumab might be seen as one of the next-generation immune checkpoint blockade drugs with a large potential; however, at the time of preparation of this manuscript, no trial was registered for camrelizumab outside of Asia.

The RATIONALE 302 trial investigated tislelizumab versus chemotherapy as second-line treatment and found that tislelizumab clinically and significantly improves the OS (median OS: 8.6 versus 6.3 months; HR 0.70; 95% CI 0.57–0.85, *p* = 0.0001) [46]. These promising novel drugs underline the potential of further investigation of immunotherapeutic agents.

## 4. Future Perspective and Discussion

It is evident that immunotherapy has changed the face of cancer treatment in recent years. In 2020 and 2021, the eagerly awaited breakthroughs that had already led to the establishment of new regimens in various cancer entities finally found their way into therapeutic strategies for esophageal cancer. The checkpoint inhibitors nivolumab and pembrolizumab have recently been approved in resectable as well as in metastatic settings, and further approvals are expected. However, several concerns in regard to immunotherapeutic agents remain.

### 4.1. Patient Selection

The first, and most obvious, concern is that not all patients profit from these therapeutic interventions. Although some studies, such as the CheckMate 577 trial, conclude that the results are similar between histological subtypes and varying expression of PD-L1, other studies show evident differences between tumor subtypes. For instance, the greater benefit for patients with SCC receiving the CROSS regimen, whether alone or in combination with nivolumab, is evident and consistent with previous knowledge. Just as evident, the better response to PD-(L)1 inhibition in tumors with high PD-L1 expression, as demonstrated in the CheckMate 649 and the KEYNOTE-181 trial, underlines the potential of PD-L1 expression irrespective of the histology.

However, the views on the implementation of such predictive markers diverge. For instance, the FDA surprised everyone with its decision on the approval of nivolumab for all patients independent of PD-L1 expression based on the CheckMate 649 trial, although the trial clearly predefined the efficacy in patients with a CPS of ≥5 as a primary endpoint and, finally, the greatest benefit could also be shown in this subgroup [27]. Although the latest update on the CheckMate 649 trial suggests some benefit for patients with a CPS of <5, longer follow-up periods will be needed in order to see whether these promising response rates will be translated into higher overall survival rates. However, the approval of pembrolizumab based on the KEYNOTE-181 trial was granted only for patients with a CPS of ≥10 [42]. Recently, the EMA restricted its approval of pembrolizumab as first-line treatment in advanced gastroesophageal cancer for patients with a CPS of ≥10, where the FDA again was broader with its decision as the approval was for all comers.

Furthermore, evidence exists that demographic factors and the expression of immunohistochemical biomarkers might be important as predictive markers similar to that of histological subtypes and tumor location themselves. The differences between Eastern and Western patients in tumor development as well as treatment response have been widely discussed for decades, yet no concise underlying mechanisms for this phenomenon could be detected [47]. Thus, the comparison of therapeutic regimens in different demographic groups is of high clinical relevance and is increasingly regarded as an important stratification factor in clinical trials. Post hoc analyses of large international studies might reveal that some demographic subgroups benefit more than others [48]. The results of the CheckMate 577 trial show a more pronounced benefit of adjuvant nivolumab treatment for white and Asian patients than for black and others (white: HR 0.71 (0.57–0.88); Asian: HR 0.70 (0.41–1.22); black: HR 0.43 (0.06–3.06); others: HR 0.48 (0.11–2.02)) [18]. Thus, a major aim for future studies is the further characterization of these cohorts.

In addition, tumor characteristics such as microsatellite instability (MSI) and tumor mutational burden (TMB) are surmised to be predictive for response to immunotherapy independent of the tumor site. Thus, in 2017 and 2020, the FDA granted the so-called tissue agnostic approvals for pembrolizumab in metastatic MSI-high and TMB-high tumors without any other satisfactory treatment options, respectively [49,50]. Yet, the expression of these markers varies vastly throughout tumor entities as well as other factors such as demography and stage [51,52,53,54]. Although these markers are only present in a small subset of patients with esophageal tumors, the survival expectancy is high when treated with immunotherapy. It is, however, a matter of discussion whether these patients could solely be treated with immunotherapy or immunotherapy combinations without a chemotherapy backbone.

As performed in other tumor entities such as lung and head and neck cancers, the evaluation of PD-L1 positivity seems to be a key element for effective treatment decisions. However, there are two major established methods to score these positive cells. The tumor proportion score (TPS), which was originally developed for lung cancer, involves the measurement of PD-L1 expression only within the tumor cells. The combined positive score (CPS), however, considers the expression of PD-L1 on tumor cells and immune cells alike. Thus, the CPS is the ratio of the number of all PD-L1-expressing cells (tumor cells, lymphocytes, macrophages) to the number of all tumor cells. Therefore, it is surmised that the CPS might paint a more accurate picture of the tumor microenvironment and, thus, is used in most large trials, even if only in post hoc analyses [55]. In the case of esophageal cancer, two major phase III trials, CheckMate 577 and 648, stratified patients based on the TPS, and the latter trial used TPS positivity even in the scope of the primary endpoint. This might cause some confusion in clinical routine since many centers have already established the CPS as a routine pathological parameter. Post hoc CPS assessment will provide further data on the potential difference of benefits for patients between CPS and TPS assessments and will provide the possibility to compare these valuable results with other clinical trials.

Another issue to be addressed when discussing recent trials is the aforementioned post hoc analyses, which provide important information about the diversity of the study population and provide new insight into potential biomarkers. However, to ensure statistical significance and patient safety, it is of great importance to specify the patient population at the time of planning the study and adjust the study design appropriately [56]. Only if the adequate study population is chosen beforehand, more accurate findings may improve our understanding of predictive markers. Oncology is, however, a rapidly changing field, and it is highly likely that a standard treatment (as seen in NEO-AGEIS) or a definition of a specific biomarker (as seen in the CheckMate 648 trial, as the study used the TPS within the primary endpoint, yet the CPS became more customary for gastroesophageal cancer patients afterward) might change during the recruitment period of a clinical trial. Since this issue seems to become increasingly frequent in the near feature, more flexible statistical designs in order to achieve reliable post hoc analyses are desirable.

As the quality of life has become an increasingly important outcome parameter, trial designs as well as therapeutic approaches are altered to maintain high standards of the quality of life as a major treatment aim. Thus, novel treatment approaches combining precision medicine with current treatment options are underway [57].

### 4.2. Overcoming Immune Cold Tumors

However, even after optimizing patient selection, the shortage of treatment options for patients without positive predictive biomarkers remains a major concern in patient management. Thus, another issue that needs to be addressed considering immunotherapy is the identification of underlying mechanisms for primary resistance to targeted therapies in patients without proper response rates. Strategies to overcome such immune cold tumors are widely researched and include several approaches [58]. A potential aim is to further neutralize immunosuppression at the tumor site by combining immunotherapeutic approaches. This approach was demonstrated in the CheckMate 648 trial by the combination of nivolumab plus ipilimumab [44]. Although the combination treatment was superior to chemotherapy alone, there was a cross-over of the curves within the Kaplan–Meier estimate, indicating that some patients were rapidly progressing under this chemotherapy-free regimen. Longer follow-up und further subgroup analyses will help to identify the specific subgroups of patients obtaining benefit from this promising chemotherapy-free regimen with nivolumab plus ipilimumab. Other trials investigating the combination of PD-(L)1 inhibitors with other immunotherapeutic approaches, such as the above-mentioned SKYSCRAPER-07 trial investigating tiragolumab or the FRACTION trial investigating the lymphocyte activation gene-3 (LAG3) inhibitor relatlimab, are underway and warrant enlightening results [59].

Furthermore, it is surmised that radiation therapy may enhance response to immunotherapeutic agents by inducing local inflammation. In addition to the highly promising findings of the CheckMate 577 trial, several other trials have investigated the combination of immunotherapeutic agents with radiation therapy. The BTCRC-ESO14-012 trial showed that adjuvant treatment with the PD-L1 antibody durvalumab improved 1-year recurrence-free survival after trimodality therapy (chemoradiation + surgery) to 79.2% compared to the historical rate of 50% [60]. Other combinations, including pembrolizumab, avelumab, and atezolizumab, are currently under investigation, and the first results are expected throughout the next few years [61,62,63].

Further strategies to overcome immune cold tumors include the modification of the tumor vasculature by targeting endothelial growth (i.e., in combination with tyrosine kinase inhibitors such as regorafenib in the REGONIVO trial or in combination with vascular endothelial growth factor receptor 2 (VEGFR-2) antibodies such as ramucirumab) as well as increasing the frequency of tumor-specific T cells with personalized approaches such as CAR T cell therapy [64,65].

### 4.3. Financial Considerations

Last but not least, a major issue when talking about novel immunotherapeutic approaches is finances. Several studies imply that the increasing financial distress due to novel, expensive treatment strategies as well as prolonged overall survival with consecutively increasing treatment costs are clinically relevant to patient-centered outcomes with a major impact on the health-related quality of life [66,67]. Although this trend has been evaluated and described for years, little has been done to effectively intervene with the problem. On the one hand, long-term solutions must be made by governments as well as over-governmental organizations such as the European Union to focus on reducing unsustainable drug prices and promoting innovative insurance models. On the other hand, for more immediate solutions, physicians as well as patients should be better educated on treatment costs, as cancer remains one of the most expensive medical conditions to treat [68].

## 5. Conclusions

The recent breakthrough in immunotherapy has revolutionized the field of esophageal cancer treatment. Although the current results of large clinical trials promise high efficacy with tolerable toxicity, desirable survival rates, and sustained quality of life, some concerns remain. Adequate patient selection, the identification of underlying mechanisms for primary resistance to immunotherapies and overcoming these, and financial toxicity pose major issues, which should be addressed in future studies. Prompt answers and solutions to these concerns will shape the treatment algorithm of gastroesophageal patients in upcoming years.

## Figures and Tables

**Table 1 cancers-13-04632-t001:** Overview of recent clinical trials discussed in this review in patients with resectable esophageal cancer.

Name	Trial Number	Tumor Type	Setting (Line)	Phase	Population	Treatment Arms	Status
CROSS	NTR487	Esophageal + GEJ carcinoma; both adenocarcinoma and squamous cell carcinoma	Neoadjuvant	III	Europe	Cohort 1: chemoradiotherapy + surgery (paclitaxel + carboplatin) Cohort 2: surgery alone	Completed
FLOT4	NCT01216644	Gastric + GEJ adenocarcinoma	Perioperative	II/III	Europe	Cohort 1: FLOT (5-FU + leucovorin + oxaliplatin + docetaxel) Cohort 2: ECF (epirubicin, cisplatin, 5-FU)	Completed
Neo-AEGIS	NCT01726452	Esophageal + GEJ adenocarcinoma	Neoadjuvant/perioperative	III	Europe	Cohort 1: perioperative chemotherapy Cohort 2: neoadjuvant chemoradiotherapy	Active, not recruiting
ESOPEC	NCT02509286	Esophageal adenocarcinoma	Neoadjuvant/perioperative	III	Europe	Cohort 1: FLOT + surgery Cohort 2: CROSS + surgery	Active, not recruiting
CheckMate 577	NCT02743494	Esophageal + GEJ carcinoma; both adenocarcinoma and squamous cell carcinoma	Post-operative adjuvant	III	North and South America, Australia, Europe, Asia	Cohort 1: nivolumab Cohort 2: placebo	Active, not recruiting
KEYNOTE-585	NCT03221426	Gastric + GEJ adenocarcinoma	Perioperative	III	North and South America, Europe, Asia	Cohort 1: pembrolizumab + chemotherapy Cohort 2: placebo + chemotherapy	Active, not recruiting
-	NCT04280822	Esophageal squamous cell carcinoma	Neoadjuvant	III	Asia	Cohort 1: chemotherapy + toripalimab Cohort 2: chemotherapy	Recruiting

**Table 2 cancers-13-04632-t002:** Overview of recent clinical trials discussed in this review in patients with unresectable or metastatic esophageal cancer.

Name	Trial Number	Tumor Type	Setting (Line)	Phase	Population	Treatment Arms	Status
KEYNOTE-975	NCT04210115	Esophageal + GEJ carcinoma	Definitive	III	North and South America, Europe, Asia	Cohort 1: definitive chemoradiotherapy (dCRT) + pembrolizumab Cohort 2: dCRT + placebo	Recruiting
SKYSCRAPER-07	NCT04543617	Esophageal squamous cell carcinoma	Post-definitive chemoradiotherapy	III	North and South America, Europe, Asia, Africa, Oceania	Cohort 1: tiragolumab + atezolizumab Cohort 2: atezolizumab + placebo Cohort 3: tiragolumab + placebo	Recruiting
CheckMate 649	NCT02872116	Gastric + GEJ + esophageal adenocarcinoma	1st line, metastatic, advanced or recurrent	III	North and South America, Europe, Asia, Australia	Cohort 1: ipilimumab + nivolumab Cohort 2: nivolumab + chemotherapy Cohort 3:chemotherapy (investigator’s choice)	Active, not recruiting
KEYNOTE-590	NCT03189719	Esophageal + GEJ carcinoma; both adenocarcinoma and squamous cell carcinoma	1st line, metastatic, advanced or recurrent	III	North and South America, Europe, Asia, Africa	Cohort 1: pembrolizumab + chemotherapy (cisplatin + 5-FU) Cohort 2: placebo + chemotherapy (cisplatin + 5-FU)	Active, not recruiting
ToGA	NCT01041404	Gastric + GEJ adenocarcinoma, HER2-positive	1st line, metastatic, advanced or recurrent	III	North and South America, Europe, Asia, Africa	Cohort 1: trastuzumab + chemotherapy (fluoropyrimidine + cisplatin) Cohort 2: chemotherapy (fluoropyrimidine + cisplatin)	Completed
KEYNOTE-811	NCT03615326	Gastric + GEJ adenocarcinoma, HER2-positive	1st line, metastatic, advanced or recurrent	III	North and South America, Europe, Asia, Oceania	Cohort 1: pembrolizumab + trastuzumab + oxaliplatin/cisplatin + capecitabine/5-FU Cohort 2: placebo + trastuzumab + oxaliplatin/cisplatin + capecitabine/5-FU	Active, not recruiting
ATTRACTION-3	NCT02569242	Esophageal squamous cell carcinoma	≥2nd line, metastatic, advanced or recurrent	III	North America, Europe, Asia	Cohort 1: nivolumab Cohort 2: chemotherapy (docetaxel or paclitaxel)	Completed
KEYNOTE-180	NCT02559687	Esophageal + GEJ carcinoma; both adenocarcinoma and squamous cell carcinoma	≥3rd line, metastatic, advanced or recurrent	II	North America, Europe, Asia	Cohort 1: pembrolizumab	Active, not recruiting
KEYNOTE-181	NCT02564263	Esophageal + GEJ carcinoma; both adenocarcinoma and squamous cell carcinoma	≥2nd line, metastatic, advanced or recurrent	III	North America, Europe, Asia	Cohort 1: pembrolizumab Cohort 2: chemotherapy (physician’s choice)	Active, not recruiting
CheckMate 648	NCT03143153	Esophageal squamous cell carcinoma	1st line, metastatic, advanced or recurrent	III	North and South America, Europe, Asia, Australia	Cohort 1: nivolumab + ipilimumab Cohort 2: nivolumab + chemotherapy (cisplatin + 5-FU) Cohort 3: chemotherapy (cisplatin + 5-FU)	Active, not recruiting
ESCORT-1st	NCT03691090	Esophageal squamous cell carcinoma	1st line, metastatic, advanced or recurrent	III	Asia	Cohort 1: camrelizumab + chemotherapy (paclitaxel + cisplatin) Cohort 2: placebo + chemotherapy (paclitaxel + cisplatin)	Recruiting
RATIONALE 302	NCT03430843	Esophageal squamous cell carcinoma	≥2nd line, metastatic, advanced or recurrent	III	North America, Europe, Asia	Cohort 1: tislelizumab Cohort 2: chemotherapy (physician’s choice)	Active, not recruiting

**Table 3 cancers-13-04632-t003:** Recent approvals of immunotherapeutic agents for esophageal cancer.

Adenocarcinoma				Squamous Cell Carcinoma			
Treatment	Trial Name	Setting	Approval	Treatment	Trial Name	Setting	Approval
Nivolumab	CHECKMATE-577	Adjuvant (after neoadjuvant chemoradiation + surgery with residual pathological disease)	2021 (FDA, EMA)	Nivolumab	CHECKMATE-577	Adjuvant (after neoadjuvant chemoradiation + surgery with residual pathological disease)	2021 (FDA, EMA)
Nivolumab + platine or fluoropyrimidine-based chemotherapy	CHECKMATE-649	1st line, metastatic, recurrent or inoperable	2021 (FDA)	Pembrolizumab + platine or fluoropyrimidine-based chemotherapy	KEYNOTE-590	1st line metastatic, recurrent or inoperable	2021 (FDA, EMA for CPS ≥ 10)
Pembrolizumab + platine or fluoropyrimidine-based chemotherapy	KEYNOTE-590	1st line, metastatic, recurrent or inoperable	2021 (FDA, EMA for CPS ≥ 10)	Nivolumab	ATTRACTION-3	2nd and further-line metastatic, recurrent or inoperable	2020 (FDA, EMA)
				Pembrolizumab	KEYNOTE-181	2nd and further-line metastatic, recurrent or inoperable, CPS ≥ 10	2019 (FDA)

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
