# Peer review of "Immunotherapy for Esophageal Cancers: What Is Practice Changing in 2021?"

_cancers, 2021, doi:10.3390/cancers13184632_

Round 1

Reviewer 1 Report

Puhr et al submitted a very well written review on current immunotherapy for esophageal cancer.   All important current data is very well presented and up to date, including 2021 congress data from ASCO.   As the authors state in the paragraph on perioperative FLOT, many patients drop out due to chemotherapy side effects and toxity in the postoperative setting. Even if the current status of immunotherapy is included, no true discussion on their reported side effects is given. I believe this would be important for the readership of cancers to have a paragraph on this topic included, and especially how to overcome typical problems. Consider citing this article in that context:   Plum PS, Damanakis A, Buschmann L, Ernst A, Datta RR, Schiffmann LM, Zander T, Fuchs H, Chon SH, Alakus H, Schröder W, Hölscher AH, Bruns CJ, Bludau M. Short-term outcome of Ivor Lewis esophagectomy following neoadjuvant chemoradiation versus perioperative chemotherapy in patients with locally advanced adenocarcinoma of the esophagus and gastroesophageal junction: a propensity score-matched analysis. J Cancer Res Clin Oncol. 2021 Jul 5. doi: 10.1007/s00432-021-03720-5. Epub ahead of print. PMID: 34223965.   Congratulations to this very nice article.    

Reviewer 2 Report

Introduction paragraph : authors should explain the geographic disparities for suamous esophageal cancer and gastro-esophageal junction (GEJ) adenocarcinoma and more specifically the incidence increase in western countries for GEJ adenocarcinoma. Incidence and mortality rates must be indicated.  Risk factors and their differences between GEJ adenocarcinoma ans squamoous histology have to be described.

Regarding tables 1 and 2 : I think that it will be better to establish 3 tables as follow : table 1 with publihed results for resectable disease ; table 2 with publihed results with unresectable locally advanced and metastatic disease ; table 3 for ongoing trials.

The tables for published trials should indicate : number of patients included, median PFS with HR, median OS with HR, eventually with PDL-1 CPS level, depending on the clinical potential impact.

line 61 : the CROSS trial included squamous and adenocarcinoma from esophagus. In the FLOT study, more than half of patients included had gastric carcinoma. The two populations from those trials were totally different. Consequently, it is impossible to compare CROSS and FLOT studies (line 75). 

Line 183 : the CheckMate 649 included a majority of gatric carcinoma. It is important to indicate the number of patients included with GEJ adenocarcinoma and the forest plt analysis.

Line 196 : In the KEYNOTE 590, results according to histology and CPS-PD-L1 have to be described and discussed.

Line 207 to 229 : authors should indicate the rate of HER 2 positive tumors (same for MSI) in GEJ adenocarcinoma compare with gastric carcinoma.

Round 2

Reviewer 2 Report

No additional comment